# Concept Wikification for COVID-19

**Panagiotis Lymperopoulos**
Tufts University
plympe01@tufts.edu

**Haoling Qiu**
Raytheon BBN Technologies
haoling.qiu@
@raytheon.com

**Bonan Min**
Raytheon BBN Technologies
& Tufts University
bonan.min@
@raytheon.com

## Abstract

Understanding scientific articles related to COVID-19 requires broad knowledge about concepts such as symptoms, diseases and medicine. Given the very large and ever-growing scientific articles related to COVID-19, it is a daunting task even for experts to recognize the large set of concepts mentioned in these articles. In this paper, we address the problem of concept wikification for COVID-19, which is to automatically recognize mentions of concepts related to COVID-19 in text and resolve them into Wikipedia titles. We develop an approach to curate a COVID-19 concept wikification dataset by mining Wikipedia text and the associated intra-Wikipedia links. We also develop an end-to-end system for concept wikification for COVID-19. Preliminary experiments show very encouraging results. Our dataset, code and pre-trained model are available at github.com/panlybero/Covid19_wikification.

## 1 Introduction

Recently, research articles about COVID-19 are published at overwhelming scale and speed. Understanding these articles is crucial for combating the COVID-19 pandemic. For example, the following sentence describes the transmission of coronavirus:

**S1**: *The interaction of the* **coronavirus spike protein** *with its complementary* **cell receptor** *is central in determining the* **tissue tropism**, **infectivity**, *and* **species range** *of the released virus. Coronaviruses mainly target* **epithelial cells**. *They are transmitted from one host to another host, depending on the coronavirus species, by either an* **aerosol**, **fomite**, *or* **fecal-oral** *route.*

There are a few concepts (in bold) that are central to understanding this sentence. Understanding these concepts requires biomedical expertise. The sheer volume of new information makes it a daunting task even for infectious disease experts to recognize the ever-growing set of concepts mentioned in the literature. This necessitates developing a concept recognition and linking system that can automatically tag mentions of these concepts in text and resolve them into Wikipedia titles.

We call this task *concept wikification for COVID-19*. It is an extension of the classic wikification (Roth et al., 2014) problem with a few new challenges introduced: first, what concepts are related to COVID-19? We need an efficient approach to identify these concepts at scale. Second, there is no labeled dataset for training such a wikifier. Therefore, we need an effective way to construct a labeled dataset timely, to assist researchers to develop concept wikifiers in time to combat the COVID-19 pandemic. Third, the concept wikifier needs to perform both concept mention detection and linking. While these two steps could be uncoupled, the developing nature of the pandemic means that new concepts and information is constantly being associated with COVID-19, and updating systems separately may inhibit timely response to the the shifting domain. This is unlike classic wikification for which many Named Entity Recognition (NER) or mention detection systems are available.

In this paper, we address all these challenges and develop a practical solution for COVID-19 concept wikification. We first develop an approach to automatically discover concepts related to COVID-19 by exploring Wikipedia, starting from the Wikipedia page for COVID-19. We then leverage Wikipedia text and intra-Wikipedia links to automatically harvest a large dataset in which mentions are labeled with their Wikipedia titles. Using this dataset, we develop an end-to-end system for concept wikification for COVID-19. Preliminary experiments show encouraging results.

Our contributions are summarized as follows:

- We developed an approach to automatically identify 7,238 concepts related to COVID-19.

- We curated a large labeled dataset for training concept wikifiers for these concepts. We made the dataset available to the public.

- We developed several neural wikifiers for COVID-19. Results are very encouraging.

## 2 Related Work

Entity Linking (EL) (Hachey et al., 2013; Guo et al., 2013; Yamada et al., 2016, 2017) or Wikification (Roth et al., 2014; Ratinov et al., 2011) is a reference resolution task, for which the goal is to identify entity mentions in text and resolve each mention into an entry in the reference knowledge base (KB) (for Wikification, the entry is a Wikipedia title). Most existing EL systems focus on entities such as persons, organizations and geo-political entities, and use NER or mention extraction systems (Ratinov and Roth, 2009) to identify candidate mentions. To perform disambiguation to Wikipedia titles, systems (Ratinov et al., 2011; Lin et al., 2017; Nguyen et al., 2016) often use lexical, syntactic and semantic features.

Recently, there has been growing interest in modeling the EL stages (e.g., mention detection, candidate generation, entity disambiguation) in a joint modeling framework such as a graphic model (Durrett and Klein, 2014) or a neural architecture (Nguyen et al., 2016; Kolitsas et al., 2018). (Broscheit, 2019) showed that a simple BERT-based token classification model can perform surprisingly well at end-to-end EL.

Entity linking research has benefit greatly from large-scale annotated datasets such as the CoNLL03/AIDA dataset (Hoffart et al., 2011). There are also approaches for specialized domains such as the bio-medical domain (Zheng et al., 2015), or in a cross-lingual setting (Sil et al., 2018; Pan et al., 2017).

This work differs from previous work in that we aim at recognizing and linking concepts related to COVID-19, to help timely and more comprehensive understanding of the concepts in the large amount of COVID-19 related scientific articles.

## 3 Building a Dataset for COVID-19 Concept Wikification

Our goal is the following: first, we aim at finding concepts that are related to COVID-19. We also avoid concepts such as persons, organizations or geo-political entities (GPE), since they are covered in existing datasets. Second, we aim at harvesting text in which mentions of these concepts are labeled with their reference Wikipedia title. These labeled texts can be used to train a concept wikifier.

We leverage Wikipedia to construct this dataset because it has broad coverage, is frequently updated by its editors and has plenty of text and intra-page links that support the goal.

### 3.1 Finding Concepts Relevant to COVID-19

We repeat the following steps until we find a sufficient number of concepts relevant to COVID-19.

- Step 1: Start with known relevant pages $\mathcal{S}$ about COVID-19: parse each page $p \in \mathcal{S}$, and then find all concepts $\mathcal{C}$ mentioned in $p$.

- Step 2: For each $c \in \mathcal{C}$, fetch the title page $p'$ for $c$ and add $p'$ into $\mathcal{S}$.

In a nutshell, we iteratively grow the set of COVID-19 relevant concepts by a breadth-first-search (BFS), starting at the Wikipedia entry for COVID-19 [1] and expanding outward by following links included in that page. We run BFS with a maximum depth of 2, to avoid exploding the search space. This results in a total of 11,795 concepts. The process is illustrated in Figure 1.

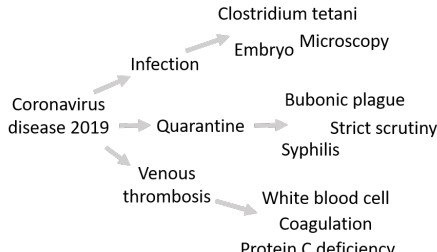

Figure 1: Sample search tree generated by a BFS starting from the Wikipedia entry for COVID-19. While only a branching factor of 3 is shown for illustration, the data collection procedure did not enforce any such limit.

Data collection from wikipedia is a noisy process, as even two iterations of BFS can reach pages that are only tangentially related to COVID-19. As a result, a few filters and post processing steps were applied to the raw data to narrow down the collection to relevant entities. First, we remove very rare concepts from the dataset, as these mostly consist of outliers. Second, we remove concepts of types

---

[1]Our initial $\mathcal{S}$ contains the single Wikipedia page about COVID-19: https://en.wikipedia.org/wiki/Coronavirus_disease_2019.

such as cities and countries, that are covered extensively in existing datasets. [2]. We also used a list of cities and countries harvested from the GeoLite2 database [3], to prune entries that have no types in WikiData. Third, a small amount of manual effort is applied to select concept types that are related to COVID-19. We then keep the concepts whose types pass through these filters.

| Dataset | # of sentences | # of concepts |
|---------|----------------|---------------|
| Full | 108,868 | 7,238 |
| Ambiguous | 61,020 | 2,636 |

Table 1: Statistics of the COVID-19 concept linking dataset. The *ambiguous* dataset only includes ambiguous mentions for which $p(c|m) < 0.9$, while the *full* dataset includes all mentions for all concepts.

### 3.2 Harvesting Labeled Mentions for Concepts

To enable training a concept wikifier, we need to harvest labeled instances where each mention (text span) of a concept is labeled with a ground-truth Wikipedia title.

We generate labeled mentions for a concept by finding its occurrences in Wikipedia pages, via embedded intra-Wikipedia links that point to the concept. These intra-Wikipedia links are provided by Wikipedia editors. Since there are an overwhelming amount of instances for concepts such as *protein* or *coronavirus*, we cap the maximum numbers of sentences per concept to 50.

**Identifying ambiguous mention strings** A mention string $m$ could be trivially resolved to its reference concept $c$ if $m$ only has one match in the vocabulary of concepts, since one could simply do string matching to resolve $m$ to $c$. It is not uncommon for a scientific term (e.g., *SARS-CoV-2*) to always link to a unique wikipedia page, as it tends to be defined without ambiguity.

Based on the raw dataset consists of all mentions of all concepts, we calculate the probability of $m$ being annotated as resolving to $c$ as $p(c|m)$, which is the count of $m$ labeled as $c$ divided by the total count of $m$. To make a more challenging dataset for training our neural wikifier, we remove all instances of $m$ where $p(c|m) \geq 0.9$ since these mention strings can be trivially resolved.

Table 1 shows the statistics of the resulting dataset. Example concept types, concepts and sentences are shown in Table 2.

## 4 End-to-End Concept Wikification

We develop a hybrid, pipelined system consists of the following two steps. Given an input sentence, it first identifies concept strings (e.g., *SARS-CoV-2*) that are unambiguous and assigns their types using

a majority-class classifier. It then applies a SciBERT (Beltagy et al., 2019)-based neural wikifier to assign types to other, more ambiguous concept mention strings. This hybrid system combines the best of both worlds: it uses the majority-class rule-based approach to resolve unambiguous concept mentions, and a deep contextualized neural model to resolve the ambiguous mentions, which require context to disambiguate.

**Majority-class classifier** We first curate a list of unambiguous concept mention strings by finding mention string $m$ such that $p(c|m) \geq 0.9$ for some $c$. It is trivial to resolve these mention strings into the corresponding concept. We perform string matching using the list of unambiguous mention strings, and simply resolve each match to its majority-class concept, which is the most frequent concept it was tagged in the dataset.

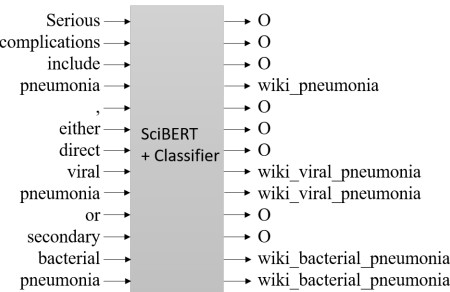

Figure 2: An illustration of the neural SciBERT-based wikifier (other models are similar). It takes a sequence of words as input and outputs a sequence of labels corresponding to wikipedia titles or *Other* (O).

**Neural BERT-based wikifier** Inspired by (Broscheit, 2019) which showed that a simple BERT-based token classification model can perform surprisingly well for end-to-end entity linking, we use a similar neural model for concept wikification: the model takes a sentence (a sequence of words) as input, passes it through the BERT (Devlin et al., 2019) architecture, and then outputs one concept name per token. To train this model for concept wikification, each sentence for train-

[2] The concept types are extracted from WikiData: `https://www.wikidata.org/wiki`.

[3] This list is available at `github.com/panlybero/Covid19_wikification`

| Concept type (%) | Example concepts | Example sentences |
|---|---|---|
| Disease (16%) | interstitial lung disease, pulmonary hypertension | Other associated lung conditions include **interstitial lung disease**, chronic diffuse **pulmonary hypertension**, pulmonary emboli |
| Infectious disease (5%) | severe acute respiratory syndrome coronavirus 2 | The coronavirus disease 2019 is an ongoing viral pandemic of **severe acute respiratory syndrome coronavirus 2** |
| Anatomical structure (4.3%) | blood vessel, capillaries | ... first to record microscopic observations of ..., spermatozoa, blood and **capillary** flow in **blood vessel** |

Table 2: Example concept types, concepts and sentences in the curated COVID-19 concept wikification dataset. The concepts span a large number of types: *disease* (16%), *infectious disease* (5%), *anatomical structure* (4.3%), *academic discipline* (3.6%), *essential medicine* (3.1%), *chemical compound* (3%), *cause of death* (3%), *medication* (2.6%), *symptom* (2%), *medical specialty* (2%).

ing is paired with a sequence of concept names. Therefore, during decoding, the model is going to output a sequence of tags consisting of either concept names or O. An example is illustrated in Figure 2. We refer readers to (Broscheit, 2019) for more details about the model. We built a few neural wikification models, one per each of the following BERT-like models:

- BERT [4] that is pretrained from the BooksCorpus and Wikipedia.

- SciBERT (Beltagy et al., 2019), a BERT variant pretrained on 1.14M scientific papers.

- BioBERT (Lee et al., 2020), another BERT variant that is pretrained on PubMed articles.

## 5 Experiments

We evaluate seven wikifiers using the *ambiguous* dataset in Table 1. Evaluating on the *full* dataset does not provide much more insight given that the additional concepts are trivial to resolve with the majority-voting string-matching baseline. The dataset was split into a training set, a development (dev) set and a test set. 60% of the data constitutes the training set, and the remaining 40% is split evenly among dev and test.

We implemented seven concept wikifiers:

- **Baseline**: a majority-class baseline simply resolves each mention (a string match of any concept) into its majority-class concept.

- **BERT**: a BERT-based wikifier, which was trained on the training portion of our curated *ambiguous* dataset. We use frozen weights from BERT and only train the token classification layer.

- **BERT-FT**: same as BERT except that we additionally finetune ("FT") BERT.

- **SciBERT**: a SciBERT-based wikifier. We use frozen weights from SciBERT and only train the token classification layer.

- **SciBERT-FT**: same as SciBERT except that we additionally finetune ("FT") SciBERT.

- **BioBERT**: a BioBERT-based wikifier. We use frozen weights from BioBERT and only train the token classification layer.

- **BioBERT-FT**: same as BioBERT except that we additionally finetune ("FT") BioBERT.

For pre-processing we use each model's built-in tokenizer to tokenize the sentences. We also use a maximum sentence length of 200 (short sentences will be padded to 200). We train all the transformer models for 50 epochs using a batch size of 200. We use the Adam optimizer with an initial learning rate $5 \times 10^{-5}$.

We use micro-average Precision (P), Recall (R) and F1 metrics, for which we only consider a concept link is true if they match the reference annotation. Similar to (Broscheit, 2019), we report two types of P/R/F1 based on whether we require strong match or weak match when calculating these metrics. Strong match requires every token in the gold annotated span to be classified correctly, while weak match only requires at least one token in the gold annotation span to be classified correctly to account for annotation inconsistencies in Wikipedia.

Experimental results in Table 3 show that the majority-class baseline is insufficient on this dataset, which indicates that this problem is challenging as many concept mentions are ambiguous and refer to more than one concepts. The BERT model performs the worst among all our models.

---

[4] https://github.com/google-research/bert

| Model | Weak | | | Strong | | |
|---|---|---|---|---|---|---|
| | P | R | F1 | P | R | F1 |
| Baseline | 0.32 | 0.80 | 0.45 | 0.32 | 0.80 | 0.45 |
| BERT | 0.89 | 0.18 | 0.30 | 0.87 | 0.14 | 0.24 |
| BERT-FT | 0.65 | 0.76 | 0.70 | 0.64 | 0.71 | 0.67 |
| SciBERT | 0.72 | 0.46 | 0.56 | 0.59 | 0.26 | 0.36 |
| SciBERT-FT | 0.67 | 0.77 | 0.72 | 0.66 | 0.73 | 0.69 |
| BioBERT | 0.75 | 0.82 | 0.79 | 0.75 | 0.81 | 0.78 |
| BioBERT-FT | 0.77 | 0.84 | **0.80** | 0.76 | 0.83 | **0.79** |

Table 3: Performances of the seven concept wikifiers on the ambiguous test dataset. **Strong** and **Weak** are strong match and weak match, respectively. For the majority-class baseline, the strong match scores are the same as the weak match scores.

This can be attributed to the significant domain mismatch between the pre-training corpus and our dataset. This is further supported by the significant performance gains that result from finetuning. SciBERT-FT and BERT-FT achieves significantly higher performance compared to the baseline and the frozen models. This shows that deep contextualized word representations need to be fine-tuned in order to work well for this challenging problem. BioBERT, the model who's domain matches the dataset the best, performs very well on the task surpassing both BERT-FT and SciBERT-FT. This supports that the specialized scope of our dataset requires contextualized word-embeddings specifically tuned to the biomedical domain. Finetuning BioBERT provides only small improvement compared to the non-finetuned version, which demonstrates that the model's pre-training domain is close enough to our dataset that the effects of additional finetuning are diminishing.

## 6 Conclusion and Future Work

In this paper, we curated a dataset to enable concept wikification research for COVID-19. We developed a few concept wikifier and shown that they perform reasonably well. As a next step, we plan to augment the dataset to include more concepts and will improve the concept wikifier using the augmented dataset.

## Acknowledgments

This work was supported by the Office of the Director of National Intelligence (ODNI), Intelligence Advanced Research Projects Activity (IARPA), via IARPA Contract No. 2019-19051600006 under the BETTER program. The views, opinions, and/or findings expressed are those of the author(s) and should not be interpreted as representing the official views or policies, either expressed or implied, of ODNI, IARPA, or the U.S. Government. The U.S. Government is authorized to reproduce and distribute reprints for governmental purposes notwithstanding any copyright annotation therein.

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
