# OpenReview forum: "Concept Wikification for COVID-19"
_EMNLP/2020/Workshop/NLP-COVID — NLP-COVID19-EMNLP Poster_

### Official Review · AnonReviewer3 · 2020-09-11
**Fine short paper**

**Rating:** 6
**Confidence:** 3

**Review:**

This paper addresses the wikification of COVID-19 concepts. Overall, I think this is a reasonable short paper for this workshop.

Quality: good enough

Clarity: good enough

Originality: OK

Significance: not life-changing but certainly an important problem.

Pros:
- The dataset seems to be adequate (it is nice that the authors release it) and the models that are applied are appropriate.
- The result that fine-tuning on the dataset improves performance is interesting.

Cons:
- It would have been nice if the authors had provided more details about the dataset, like the distribution of concepts or a list of the most common concepts.
- The strong match SciBERT (no fine-tune) seems to perform worse than the baseline. Why? Some error analysis here would be helpful. Does this suggest that F-score is not the best metric here? There seems to be a big difference between precision and recall for the baseline.
- As for the evaluation, it would have been an interesting baseline to compare SciBERT with BERT-base, although this omission does not warrant a rejection in my opinion. Please include it if you're able to run the experiment.

---

> ### Author Response · Authors · 2020-09-27
> **Response to review 3**
>
> Thank you for your thoughtful and detailed feedback.
>
> We will provide a detailed description of the dataset in the camera-ready version, as well as in the Github repository of this dataset. We opted to limit such a description of the dataset to tables 1 and 2, given the page limit for short paper submission.
>
> As you pointed out, the performance of the baseline is mostly driven by its high recall. We believe this should be expected, given the nature of the baseline method. Specifically, the baseline approach performs string matching and assigns to matches the majority concept for that string. As a result, the baseline approach will always detect spans that appear in the training set and will link them to their majority concept. Therefore, the baseline approach never predicts "O" for spans seen in the training set, regardless of the context of the mention. As a result, it produces few false negatives yielding high recall but also many false positives, yielding low precision. At the same time, the SciBERT model, while it performs better in the weak metrics, is not effective at identifying the entire mention-span correctly without any fine-tuning. As a result, it suffers in performance.
>
> Finally, we experimented with BERT-base as well in our initial experiments. While we do not at the moment have specific metrics to report for that model, the performance of BERT-base was significantly worse than that of SciBERT. As a result, we decided to focus our efforts on SciBERT instead. We attribute that to the difference in training data of the two models, as the training data for SciBERT consists of scientific text, which is closer to the dataset that we are creating with this work. We can include a comparison with using BERT-base in the camera-ready version.

---

### Official Review · AnonReviewer1 · 2020-09-12
**Adequate contribution but problems regarding the reported results**

**Rating:** 7
**Confidence:** 3

**Review:**

This paper presents a new dataset of COVID-19-related mentions extracted from Wikipedia, and some experiments for concept mapping and entity linking. The dataset was annotated by using the available curated links to COVID-19-related concepts, and authors conducted some postprocessing to remove annotations of non-medical mentions (e.g. cities, organizations or persons). This dataset was then used as an experimental resource to test entity linking experiments, where concepts are restricted to the COVID-19 disease and occur in Wikipedia ("Concept wikification"). The authors applied a 2-step approach for the task: first, unambiguous entities are string-matched, then ambiguous entities are resolved through 3 methods: a majority rule-based voting scheme, a BERT-based classifier using SciBERT, and a SciBERT-based classifier fine-tuned on the dataset. By using this latter approach, the authors obtained promising results.

I liked the article and I think could provide an interesting contribution to the workshop, especially owing to the release of this annotated dataset and the trained models.

The weaknesses I found are related to the experimental methods: because results were only reported on one round of experiments, the generalizability of the outcomes are not solid enough. A good methodological approach is to test the model in several rounds with different random seeds or random initializations, and then report the average F-score and standard deviation.

Finally, I think a more thorough description of the dataset is missing (e.g. number of tokens, examples of entity types...).
I provide some other comments below.

-Sect. 4: authors could also take a look at BioBERT, which was trained on PubMed article abstracts
Lee J, Yoon W, Kim S, et al. BioBERT: a pre-trained biomedical language representation model for biomedical text mining. Bioinformatics. 2020;36(4):1234-1240. doi:10.1093/bioinformatics/btz682

-Sect. 5, Table 2: What does "essential medicine" stand for? General medical concepts? Generic or frequent medication names? Please, explain this or give an example.

-Others (grammar, style...):

P. 2, Sect. 2. parag. 1: References need to be sorted: e.g. "(Ratinov et al., 2011; Lin et al., 2017; Nguyen et al., 2016)" => "(Ratinov et al., 2011; Nguyen et al., 2016; Lin et al., 2017)". Please, check throughout the article

P. 4: "Marjority" -> Majority

---

> ### Author Response · Authors · 2020-09-27
> **Response to review 1**
>
> Thank you for your thoughtful and detailed feedback.
>
> While we believe that the results reported in the submitted version are representative of the performance that each of our models achieves in the test set, we will address this concern in the camera-ready version by providing results averaged over many runs, along with standard deviation.
>
> We will provide a detailed description of the dataset in the camera-ready version, as well as in the Github repository of this dataset. We opted to limit such a description of the dataset to tables 1 and 2, given the page limit for short paper submission.
>
> Thanks for the suggestion on BioBERT. We would be happy to add experiments using BioBERT in the camera-ready version.
>
> According to the WHO, “Essential medicines are those that satisfy the priority health care needs of the population.” As noted in the description for table 2, 3.1% of the concepts of this type appear in our dataset. A common example of such an essential medicine is hand sanitizer.
>
> We will make the grammar and style changes suggested by the reviewer.

---

### Official Review · AnonReviewer2 · 2020-09-25
**A good short paper that could benefit a lot by adding more analysis.**

**Rating:** 7
**Confidence:** 3

**Review:**

This paper proposes an end-to-end phrase identification and linking model for COVID-19 concepts. Experiment results look pretty good in terms of F1. Some more explanation and analysis would substantially benefit this paper.

1. One limitation is that this paper over-emphasized the necessity of being end-to-end. It says "... the concept wikifier has to perform ... in an end-to-end fasion,..." No, it doesn't have to. Being end-to-end only makes it look nice.

2. Experiment results show SciBERT has terrible Recall. This might be due to domain issue. But why is this also the case for SciBERT-FT? Maybe there are some limitations in the process/model that overkills candidates, but no explanation/analysis on this.

---

> ### Author Response · Authors · 2020-09-27
> **Response to review 2**
>
> Thank you for your thoughtful and helpful feedback.
>
> We agree that the concept wikifier does not have to be end-to-end, as long as it performs well in this task. We will make adjustments to this point so that we won't over-emphasize the necessity of being end-to-end in the camera-ready version.
>
> Table 3 shows that the recall for SciBERT-FT (0.77 and 0.73 for the weak and strong recall respectively) is decent and is also significantly higher than SciBERT (0.46 weak recall and 0.26 strong recall). This shows that the fine-tuning mitigates the possible domain mismatch. We will add more explanation and analysis into the final version.